# Differences in Characteristics, Hospital Care, and Outcomes between Acute Critically Ill Emergency Department Patients Receiving Palliative Care and Usual Care

**DOI:** 10.3390/ijerph182312546

**Published:** 2021-11-28

**Authors:** Julia Chia-Yu Chang, Che Yang, Li-Ling Lai, Hsien-Hao Huang, Shih-Hung Tsai, Teh-Fu Hsu, David Hung-Tsang Yen

**Affiliations:** 1Department of Emergency Medicine, Taipei Veterans General Hospital, Taipei 11217, Taiwan; juliahazard@hotmail.com (J.C.-Y.C.); hhhuang@vghtpe.gov.tw (H.-H.H.); tfhsu@vghtpe.gov.tw (T.-F.H.); 2School of Medicine, National Yang Ming Chiao Tung University, Taipei 11221, Taiwan; 3Department of Nursing, Taipei Veterans General Hospital, Taipei 11217, Taiwan; cyang@vghtpe.gov.tw (C.Y.); lllai@vghtpe.gov.tw (L.-L.L.); 4Institute of Emergency and Critical Care Medicine, College of Medicine, National Yang Ming Chiao Tung University, Taipei 11221, Taiwan; 5Department of Emergency Medicine, National Defense Medical Center, Taipei 11490, Taiwan; tsaishihung@yahoo.com.tw; 6Department of Nursing, Yuanpei University of Medical Technology, Hsinchu 30015, Taiwan

**Keywords:** emergency department, end-of-life care, palliative care

## Abstract

Background: The early integration of palliative care in the emergency department (ED-PC) provides several benefits, including improved quality of life with optimal comfort measures, and symptom control. Whether palliative care could affect the intensive care unit admissions, hospital care and resource utilization requires further investigation. Aim: To determine the differences in inpatient characteristics, hospital care, survival, and resource utilization between patients receiving palliative care (ED-PC) and usual care (UC). Design: Retrospective observational study. Setting/participants: We enrolled consecutive, acute, critically ill patients admitted to the emergency intensive care unit at Taipei Veterans General Hospital from 1 February 2018 to 31 January 2020. Results: A total of 1273 patients were evaluated for unmet palliative care needs; 685 patients received ED-PC and 588 received UC. The palliative care patients were more severely frail (AOR 2.217 (1.295–3.797), *p* = 0.004), had functional deterioration with three ADLs (AOR 1.348 (1.040–1.748), *p* = 0.024), biopsychosocial discomfort (AOR 1.696 (1.315–2.187), *p* < 0.001), higher Taiwan Triage and Acuity Scale 1 (*p* = 0.024), higher in-hospital mortality (AOR 1.983 (1.540–2.555), *p* < 0.001), were four times more likely to sign an DNR (AOR 4.536 (2.522–8.158), *p* < 0.001), and were twice as likely to sign an DNR at admission (AOR 2.1331.619–2.811), *p* < 0.001). Palliative care patients received less epinephrine (AOR 0.424 (0.265–0.678), *p* < 0.001), more frequent withdrawal of an endotracheal tube (AOR 8.780 (1.122–68.720), *p* = 0.038), and more narcotics (AOR1.675 (1.132–2.477), *p* = 0.010). Palliative care patients exhibited lower 7-day, 30-day, and 90-day survival rates (*p* < 0.001). There was no significant difference in the hospital length of stay (LOS) (21.2 ± 26.6 vs. 21.7 ± 20.6, *p* = 0.709) nor total hospital expenses (293,169 ± 350,043 vs. 294,161 ± 315,275, *p* = 0.958). Conclusion: Acute critically ill patients receiving palliative care were more frail, more critical, and had higher in-hospital mortality. Palliative care patients received less epinephrine, more endotracheal extubation, and more narcotics. There was no difference in the hospital LOS or hospital costs between the palliative and usual care groups. The synthesis of ED-PC is new but achievable with potential benefits to align care with patient goals.

## 1. Introduction

The ED is the port of entry providing care for acute critically ill patients with serious life-limiting illnesses. The ED is designed with the purpose of saving lives at all costs; hence, the clinical paradigm continues to focus on the treatment of acute illness and injury. However, aggressive life-sustaining and disease-directed treatments in the ED may not align with the treatment goals of all patients, especially those with advanced end-stage diseases. Palliative care (PC), on the other hand, may be a better alternative but is not often addressed in the busy ED setting. PC, as defined by the World Health Organization, is “an approach that improves the quality of life of patients and their families facing the problem associated with life-threatening illness, through the prevention and relief of suffering by means of early identification, impeccable assessment, and treatment of pain and other problems, physical, psychosocial and spiritual” [1]. Since ED often establishes the in-hospital trajectory of care for patients and seriously ill, older adults in an urban ED have substantial palliative care needs [2], the integration and early implementation of palliative care in the ED has become increasingly important. Palliative care was demonstrated to decrease the intensive care unit (ICU) admissions [3], decrease the inpatient hospital length of stay (LOS) [4], and decrease the costs [5]. Despite mounting interest in integrating palliative care in the ED, little emphasis has been placed on delivering goal-concordant palliative care in most ED; additionally, palliative consultation is frequently carried out late after admission to a hospital. The dominant paradigm of hospital care in the ED placed emphasis on maintaining life at all costs, often without attention to a patient’s prognosis, treatment values, and preferences for care. This is notable since the majority of older adults with serious illnesses report that they prefer medical therapies and minimize the experience of pain and other burdensome symptoms [6]. However, complete resuscitative efforts in the face of immediate crises are often initiated in the ED due to uncertainty, time constraints [7], and medical–legal concerns [8]. These aggressive resuscitative measures would seem misaligned with patient preferences, painfully and unnecessarily prolonging the dying process in situations that are clearly futile [9]. ED frequently cares for patients at the EOL or with life-limiting illnesses, who may benefit from palliative care, which creates an invaluable opportunity to align the care trajectory with patient goals.

Our study aims to determine if the early integration of palliative care in the ED resulted in a difference in the hospital care, LOS, and mortality between the patients receiving palliative care and those receiving usual care.

## 2. Methods

### 2.1. Design

We conducted a retrospective observational analysis in the ED of a tertiary medical center. The Taipei Veterans General Hospital (TVGH) Institutional Research Board approved this study and waived the need for patient consent (2020-11-010BC).

### 2.2. Setting

TVGH, a 3000-bed university-affiliated medical center, conducted an annual ED census (85,182 ± 1821) (mean ± standard deviation (SD)) over the past five years. The emergency ICU (EICU) is a 13-bed ICU within the ED, where acutely and critically ill patients who are not admitted to the specialized ICU immediately after initial ED resuscitation and stabilization receive intensive care.

### 2.3. Participants

Figure 1 shows a flow chart of subject selection. The evaluation of the unmet needs of PC was initiated for acutely and critically ill patients aged ≥18 years who were admitted to the EICU between 1 February 2018 and 31 January 2020. Those patients aged <18 years and those with medical records containing incomplete or missing data were excluded.

### 2.4. Protocol

A total of 2814 acute critically ill patients were evaluated for the unmet needs of PC with 1541 patients excluded because PC was not required. Among a total of 1273 patients with unmet needs of palliative care, 685 (53.8%) patients and their families consented to receive ED-PC, while 588 (46.2%) patients received UC (Figure 1). Two trained authors blindly entered the abstract data for the study analyses.

### 2.5. PC Assessment

The utilization criteria were formulated by PC and hospice specialists and were adopted to identify the patients at a high risk of poor clinical outcomes, whose care commonly involved the prolonged use of advanced medical resources or technologies.

### 2.6. Outcome Measures

The primary outcomes of the studied patients were in-hospital mortality and end-of-life (EOL) care. The EOL included endotracheal intubation and ventilator support, cardiopulmonary resuscitation (CPR), cardioversion/defibrillation, epinephrine, vasopressor, cardiac pacemaker, ECMO, endotracheal removal, and narcotics. The secondary outcomes included the clinical characteristics and healthcare utilization.

### 2.7. Data Analysis

The data are expressed as the mean ± SD for continuous variables and as a number (%) for categorical variables. The data distribution was assessed using the Kolmogorov–Smirnov test. Numerical variables were compared using an unpaired *t*-test (parametric data) or the Mann–Whitney *U* test (non-parametric data). Categorical variables were compared using a two-sided chi-square or Fisher’s exact test. The factors associated with primary and secondary outcomes, respectively, showing statistical significance (*p* < 0.05) in the univariate analysis were included in the multiple regression analysis. The survival time was calculated from the date of admission to the date of death in index hospitalization using the Kaplan–Meier method; additionally, the difference in the survival time between the ED-PC and UC groups was compared using the log-rank test. The statistical significance was set at *p* < 0.05. Statistical analysis was performed using SPSS software (version 22.0; SPSS Inc., Chicago, IL, USA).

## 3. Results

A total of 1273 patients were evaluated for unmet PC needs. A total of 685 patients agreed to participate in ED-PC, while 588 patients received the UC (Figure 1). Table 1 shows the data in terms of acute critical and life-limiting illness (item A); 49.5% had septic shock, ARDS, multiple organ failure, or impending death (A7); 47.8% had advanced central neurological diseases (long-term bed-bound) combined with repeatedly or severely progressive deterioration or recurrent pneumonia, shortness of breath or respiratory failure requiring hospital admission (A6); and 26.9% of the patients had advanced cancer, metastatic, or locally aggressive disease (A1). The patients receiving ED-PC compared to UC had more advanced cancer (30.7% vs. 22.4%, *p* = 0.001), fewer advanced central neurological disease in long-term bed-bound patients (43.6% vs. 52.6%, *p* = 0.002), and more severe frailty (9.3% vs. 4.4%, *p* = 0.001).

Table 2 shows the clinical characteristics between patients receiving ED-PC and UC. The mean age of the study patients was 82.5 ± 13.7 y/o; patients receiving palliative care were younger than those receiving usual care (81.7 ± 14.3 y/o vs. 83.4 ± 13.0, *p* = 0.024). Notably, 35.2% of the study patients were women, 78.4% of the patients lived with family, and 534 (41.9%) patients had in-hospital mortality. There was no significant difference in age, sex, insurance status, living conditions, marital status, religion, educational level, Charlson comorbidity index, Acute Physiology and Chronic Health Evaluation (APACHE) II score at admission, hospital LOS, and the total hospital expense between patients who underwent palliative care or usual care. The patients receiving ED-PC were more often triaged under Taiwan Triage and Acuity Scale (TTAS) 1 (42.8% vs. 33.8%, *p* = 0.002), had higher GCS 3-4 (17.1% vs. 13.1%, *p* = 0.035), and higher in-hospital mortality (52.8% vs. 29.3%, *p* < 0.001). There was no significant difference in the hospital LOS (21.2 ± 26.6 vs. 21.7 ± 20.6, *p* = 0.709) and total hospital expense (293,169 ± 350,043 vs. 294,161 ± 315,275, *p* = 0.958) between palliative care and usual care. A total of 1151 (90.4%) patients signed a DNR, among whom 668 patients had ED-PC and 483 patients had UC (97.5% vs. 82.1%, *p* < 0.001). Among 827 (65.01%) patients who signed a DNR at admission, 533 patients had palliative care, and 294 patients had usual care (77.8% vs. 50.0%, *p* < 0.001).

Table 3 shows the multiple logistic regression analysis of the clinical characteristics between the patients receiving palliative care and usual care. Patients receiving palliative care had greater odds of being very severely frail (A8, AOR 2.217 (1.295–3.797), *p* = 0.004), appearing to have progressive functional deterioration with ≥ADLs requiring assistance (B2, AOR 1.348 (1.040–1.748), *p* = 0.024), biopsychosocial discomfort (B3, AOR 1.696 (1.315–2.187), *p* < 0.001), and in-hospital mortality (AOR 1.983 (1.540–2.555), *p* < 0.001). ED-PC patients had higher odds of TTAS 1 and fewer TTAS 3 (AOR 0.649 (0.470–0.896), *p* = 0.024). Patients receiving ED-PC were 4.5 times more likely to sign DNR forms (AOR 4.536 (2.522–8.158) *p* = 0.001) and 2.1 times more likely to sign DNR forms at admission (AOR 2.133 (1.619–2.811) *p* = 0.001).

Table 4 compared the end-of-life care in hospitalization between patients with mortality under palliative care and usual care. It was observed that more palliative care patients died in the hospice unit (16.6% vs. 7.6%), fewer in the ICU (32.6% vs. 36.0%), and fewer in ordinary wards (40.1% vs. 47.1%, *p* = 0.030). Palliative care patients received less epinephrine (15.5% vs. 32.6%, *p* < 0.001), more cases with withdrawal of the endotracheal tube (4.4% vs. 0.6%, *p* = 0.018), and more administration of narcotics (61.6% vs. 44.2%, *p* < 0.001). There was no significant difference in the endotracheal intubation, CPR, cardioversion or defibrillation, vasopressors, cardiac pacemaker, ventilator support, and extracorporeal membrane oxygenation (ECMO) or intra-aortic balloon pump (IABP). The patients with mortality under palliative care had higher DNR orders compared to usual care (99.2% vs. 95.9%, *p* = 0.010). There was no significant difference in the numbers of DNR signed at admission between palliative care and usual care patients with mortality (78.7% vs. 72.1%, *p* = 0.091).

Table 5 shows the multiple logistic regression analysis on EOL care in hospitalization between the patients receiving palliative care and usual care. The patients receiving palliative care received less epinephrine (AOR 0.424 (0.265–0.678), *p* < 0.001), more instances of withdrawal of the endotracheal tube (AOR 8.780 (1.122–68.720), *p* = 0.038), and additional narcotics (AOR 1.675 (1.132–2.477), *p* = 0.010). Figure 2 shows the survival curve of patients receiving palliative care and usual care. The patients receiving palliative care had lower 7-day, 20-day and 90-day survival (*p* < 0.01).

## 4. Discussion

This study found that patients receiving palliative care had higher odds of being severely frail, exhibiting progressive functional deterioration with ≥three ADLs requiring assistance, experiencing biopsychosocial discomfort, and higher in-hospital mortality. The patients receiving palliative care exhibited higher odds of triage under TTAS 1 compared to those with TTAS 3. Our finding is in accordance with other studies where seriously ill, older adults in an urban ED have substantial PC needs [2].

### 4.1. Hospital Care

Traditionally, the dominant paradigm of hospital care in the ED placed emphasis on maintaining life at all costs, often without attention to a patient’s prognosis, treatment values, or preferences for care. Our study found that an early integration of palliative care in the ED resulted in a difference in hospital care between palliative care and usual care patients. Palliative care patients received less epinephrine, more often experienced withdrawal of ET tubes, and were administered more narcotics. However, there was no significant difference in the endoctracheal intubation, CPR, cardioversion or defibrillation, vasopressors, cardiac pacemaker, ventilator support, or ECMO/IABP. Our finding is different from studies where a high number of patients who received early palliative care had less aggressive care at the EOL. In our study, the palliative patients had just as regular treatments with the only difference in decreased epinephrine and more endoctracheal extubation. Appropriate decisions for palliative care are dependent on accurately prognosticating a patient’s disease process and predicting impending mortality; this in itself can be challenging. Moreover, transitioning from active resuscitation to palliative treatment requires good communication between physicians and patients regarding the concept of futility. However, life-threatening emergencies usually do not allow for in-depth discussions. The fact that medical futility has no uniform definition poses another challenge. As a result, some patients at EOL experience a dying process that does not comply with the basic understanding of a “good death” [10] In fact, patients, their family, and surrogates were often observed to be dissatisfied with the hospital care provided to dying patients, who often experienced severe pain, dysphoria, and fatigue, and who underwent life-sustaining treatments [11] This might explain our finding that despite palliative implementation, we failed to deliver less aggressive hospital care with different yet similar resuscitative measures between palliative care and usual care.

However, decision-making regarding hospital care is a continuous process. The decision for full resuscitation in the face of an emergency may transition to EOL care when medical futility is evident. This is reflected in our study with a similar number of endoctracheal intubations between palliative care and usual care, but a higher number of endoctracheal extubations among palliative patients. These findings indicate that the challenge for emergency physicians is the transition from resuscitative care to palliative care. The importance and difficulty of ED as an ideal place for conversations on withholding resuscitative efforts to minimize futile treatment is also highlighted. As these EOL discussions increasingly fall within the scope of emergency medicine practice, more EPs need to be educated about and comfortable with palliative care. Studies have shown that late referral to palliative care is inadequate to alter the quality and delivery of care provided to patients with cancer [12] Therefore, in order to have a meaningful impact on the patient’s quality of and expectations for care, palliative services must be initiated as early as upon arrival to the ED. The study showed that the integration of palliative care in the ED is the first step forward. However, a significant amount of work is still needed, and many obstacles must be overcome for the introduction of palliative care to the current culture of emergency medicine. Hence, it is paramount that EPs recognize medical futility in the face of eventual death and initiate conversations on palliative and EOL care. The transition from curative care to palliative care may appropriately “help patients and their families achieve greater control over the dying process by improving the EOL care” [13].

### 4.2. Narcotic Use

Opioids are mainstay treatments for dyspnea in palliative care because they diminish respiratory drive in response to hypoxia and hypercapnia [14]. Opioids have the added benefit of treating pain and anxiety for patients suffering from dyspnea [15]. Our study showed that palliative care patients received more narcotics than usual care patients. By integrating disease-specific treatment with more aggressive strategies of symptom management and realistic goal-setting and communication, palliative medicine lessens avoidable suffering and maximizes the quality of life. However, significant barriers exist concerning the appropriate use of opioids during EOL care [16], particularly so in older patients [17]. The suboptimal prescription of opioids is due to knowledge deficits, attitudinal concerns, and unfounded perceptions of opiates hastening death [18]. Physicians should be made aware that patients often visit the ED because of new or worsening symptoms, and that EOL symptoms can be extremely distressing and at times unbearable. Our study demonstrated that nearly 56% (299/534) of patients received narcotics during EOL care. One study found that 50 percent of conscious patients who died in the hospital experienced moderate-to-severe pain in their last days [19]. Thus, it is especially important that pain, dyspnea, and other symptoms at EOL care should be properly addressed.

### 4.3. Place of Death

The results of this study show that more palliative care patients died in hospice units, fewer in ICUs, and fewer in ordinary wards. This is consistent with another study wherein the palliative care during terminal hospitalization was associated with lower ICU admission [20]. A gap exists in the delivery of goal-concordant care, with steadily increasing ICU admissions from the ED despite the fact that some patients with serious illnesses would prefer a ward or hospice, not to mention that the ICU admission admissions near the EOL are associated with worse quality of life for patients [21]. Emergency providers play a key role as they set the trajectory for patient care, including whether the patient is hospitalized and to which setting. Our study agreed with previous findings that palliative care patients are less likely to be admitted to the ICU [20]. However, not all palliative care patients actively refuse ICU admission. Hence, the early initiation of palliative care in the ED can better devise treatment plans and patient disposition to match the patients’ preferred goals and the place of care. A failure to have these conversations ensures that some patients will experience unwanted ICU admissions. Our study showed that ED-PC resulted in better concordance with the patients’ wishes with more referral to hospice care, ordinary wards, and less ICU admission.

### 4.4. Mortality

Patients receiving palliative care had lower 7-day, 30-day, and 90-day survival rates and 1.983 odds ratios of in-hospital mortality. Although receiving similar hospital care with a difference of decreased epinephrine and additional endotracheal extubation, the patients receiving palliative care had lower survival and higher mortality rates. Our study demonstrated that despite receiving similar hospital care with the difference in a decrease in epinephrine and more frequent endotracheal extubation, the palliative care patients still succumbed to lower survival. One might argue that delivering aggressive resuscitative measures, such as endotracheal intubation, CPR, or cardioversion, in critically ill patients with terminal disease who prefer palliative care may prove futile and only add to unnecessary suffering. We hypothesized that the patient characteristics upon arrival at the ED may influence the patient survival. Among our study participants, 49.5% had septic shock, ARDS, multiple organ failure, or impending death; 47.8% had advanced central neurological disease (long-term bed-bound) combined with repeatedly or severely progressive deterioration or recurrent pneumonia, shortness of breath or respiratory failure requiring hospital admission, and 26.9% had advanced cancer, metastatic or locally aggressive disease (Table 1). Our study results demonstrated that multifactorial clinical characteristics are associated with earlier and higher mortality in patients receiving palliative care and require further investigation.

### 4.5. LOS and Hospital Costs

Our study found no differences in the hospital LOS or total hospital expenses between patients receiving ED-PC and UC. Our study is in discord with studies wherein palliative care entailed significantly lower total direct and ancillary costs per day compared to usual care [20]. One study found that palliative care for advanced disease was associated with significantly lower direct hospital costs, including the costs for pharmacy, nursing, laboratory, and radiology compared to the costs for usual care patients with advanced disease [6]. The cost reduction is due to the discussion of treatment resulting in lesser use of tests, inappropriate technology, and the ICU [20].

However, our findings agree with Emanuel et al. [22] that the efforts to improve the EOL care are not necessarily cost saving. High-quality palliative care, including medication and therapy for pain relief and assistance with activities of daily living, requires additional skilled and sometimes costly caregiver support [22]. Penrod JD et al. also found that palliative care may lead to an added cost for pain and other palliative medications [5]. Although ICU costs account for approximately 20% of the overall hospital costs [20], our study showed that a reduction in ICU admission among ED-PC patients did not result in a reduction in the total hospital expenses. Wu et al. found that an early initiation of palliative care consultation in the ED was associated with a significantly shorter LOS by 3.6 days for patients admitted to the hospital [4]. Reyes-Ortiz et al. also found that patients with life-limiting illnesses seen by palliative care teams in EDs experienced a decrease in the inpatient LOS [23]. Our results, on the other hand, concurred with Penrod JD et al., that there was no significant difference in the LOS between ED-PC and UC patients [20]. One important role of palliative care is to help transition aggressive and futile efforts of prolonging life to permit a comfortable and dignified death. In this respect, palliative care has a value that is not adequately captured either in the LOS or cost of services [24], especially when there is no true good measurement for valuing the quality of death.

### 4.6. Difficulty of ED-PC

Although there is a growing realization that patient treatment goals may not align well with the traditional emergency medicine paradigm, and that many older adults presenting to the ED have substantial unmet palliative care needs [25], there are still several barriers to integrating palliative care into the ED. The emergency providers have previously identified time constraints and implementation logistics as the most challenging limitations to providing palliative care services in the ED [26]. The patient problems may range from simple titration of pain medication to complex and lengthy discussions on the goals of care with family members, who are in disagreement or denial. The majority of the emergency physicians lack the time necessary to provide in-depth discussion and would welcome the incorporation of palliative care teams in the ED, which could be cost- and labor-intensive. Moreover, initiating EOL talks in a pressured ED environment can appear ill-timed and insensitive. Nonetheless, the care models should be developed and implemented to incorporate the appropriate knowledge skills and attitudes toward palliative medicine into the ED to align the care trajectory with patient goals.

This study has several limitations. First, as a retrospective observational study, it was subject to missing or incomplete data. Second, although the inclusion criteria were strictly followed, there may still be confounding discrepancies between the criteria and clinical conditions of the patients recruited. Third, although the study determined the differences in hospital care and the outcomes between ED-PC and UC, it did not stratify the patients into subgroups based on diagnoses, such as intracranial hemorrhage, out-of-hospital cardiac arrest, community or healthcare acquired pneumonia, etc. Fourth, the study determined the patient characteristics associated with receiving ED-PC; however, the rationale for choosing ED-PC and UC remains unclear in these patients. Fifth, the study did not assess the psychosocial aspects, such as patient and surrogate viewpoints, their satisfaction or dissatisfaction, and the patients’ quality of death associated with receiving ED-PC vs. UC.

## 5. Conclusions

Our study found that acute critically ill patients receiving palliative care were more frail, more critical, and had higher in-hospital mortality rates. Palliative care patients received less epinephrine, more frequent endotracheal extubation, and more narcotics. There was no difference in the hospital LOS or hospital costs between the palliative care and usual care groups. The synthesis of ED-PC is a new but achievable concept with potential benefits to align care trajectory with patient goals.

## Figures and Tables

**Figure 1 ijerph-18-12546-f001:**
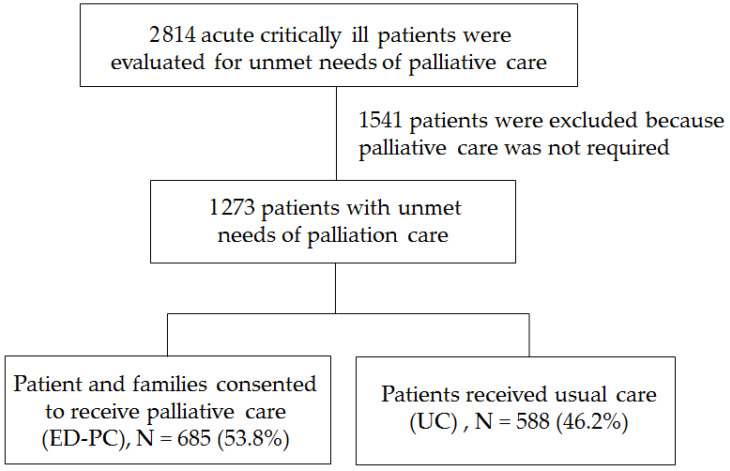
Flowchart of study patients receiving palliative care or usual care.

**Figure 2 ijerph-18-12546-f002:**
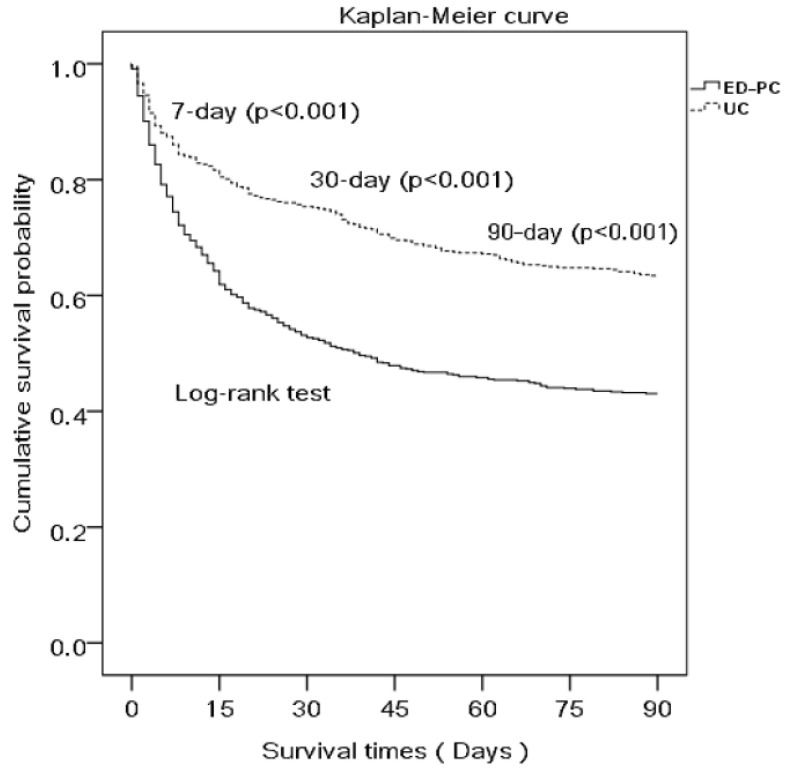
The survival curve of patients with ED-PC and UC.

**Table 1 ijerph-18-12546-t001:** The unmet needs assessment of 1273 patients with palliative care in the ED (ED-PC) and usual care (UC) at the time of admission.

Items	Overall *n* = 1273 (%)	ED-PC *n* = 685 (%)	UC *n* = 588 (%)	*p*
Acute critical and life-limiting illness				
1. Advanced cancer, metastatic, or locally aggressive disease *	342 (26.9)	210 (30.7)	132 (22.4)	0.001
2. Advanced COPD needing long-term oxygen therapy or respiratory failure requiring assisted ventilation	29 (2.3)	15 (2.2)	14 (2.4)	0.545
3. End-stage liver disease, e.g., cirrhosis, that repeatedly appears with jaundice, ascites, peritonitis, hepatic coma, esophageal varices	31 (2.4)	20 (2.9)	11 (1.9)	0.226
4. Acute or chronic renal failure, decision of not receiving dialysis	42 (3.3)	14 (2.0)	28 (4.8)	0.089
5 Advanced cardiovascular diseases (chronic heart failure NYHA III or IV, chest pain, or dyspnea while in minimal exercise or exertion, or devastating inoperable peripheral vascular diseases)	126 (9.9)	63 (9.2)	63 (10.7)	0.366
6. Advanced central neurological diseases (e.g., stroke, dementia) (long-term bed-bound) combined with repeatedly or severely progressive deterioration or recurrent pneumonia, shortness of breath, or respiratory failure requiring hospital admission	608 (47.8)	299 (43.6)	309 (52.6)	0.002
7. Septic shock, ARDS, multiple organ failure, or impending death (other devastating diseases)	630 (49.5)	334 (48.8)	296 (50.3)	0.574
8. Very severely frail (completely dependent, approaching the end-of-life, CSHA-CFS > scale 8 and 9) *	90 (7.1)	64 (9.3)	26 (4.4)	0.001
B. The unmet palliative care needs				
1. Medical care staffs would not be surprised if the patient died within 12 months of this episode	1016 (79.8)	547 (79.9)	469 (79.8)	0.967
2. Appearance of progressive functional deterioration with ≥ three ADLs needing for assistance *	621 (48.8)	360 (52.6)	261 (44.4)	0.004
3. Appearance biopsychosocial discomforts needing hospital admission *	668 (52.5)	392 (57.2)	276 (46.9)	<0.001
4. Patients with three or more unexpected emergency department visits or hospital admissions within 6 months, with symptoms consistent with a terminal or degenerative chronic medical condition	536 (42.1)	288 (42.0)	248 (42.2)	0.962
5. Patients weight loss 10% or BMI ≦ 18 within 6 months	16 (1.3)	12 (1.8)	4 (0.7)	0.087
6. Bed-bound patients with long-term unhealed bed sore or ulceration*	142 (11.2)	61 (8.9)	81 (13.8)	0.006
7. Needing complicated medical care and assistance of medical decisions, including do-not-resuscitate order, ventilator, or nutritional support	1173 (92.1)	635 (92.7)	538 (91.5)	0.426
8. Patient’s family request of palliative care *	73 (5.7)	50 (7.3)	23 (3.9)	0.010

COPD = chronic obstructive pulmonary disease; NYHA = New York Heart Association; ARDS = adult respiratory distress syndrome; CSHA-CFS = Chinese-Canadian study of health and aging clinical frailty scale; ADL = activities of daily living; BMI = body mass index. * *p* < 0.05, considered statistically significant using chi-squared analysis.

**Table 2 ijerph-18-12546-t002:** Comparison of the clinical characteristics between patients with ED-PC and UC.

Variables	Overall *n* = 1273 (%)	ED-PC *n* = 685 (%)	UC *n* = 588 (%)	*p*
Age (y) *	82.5 ± 13.7	81.7 ± 14.3	83.4 ± 13.0	0.024
<65	165 (13.0)	104 (15.2)	61 (10.4)	0.072
65–75	138 (10.8)	76 (11.1)	62 (10.5)	
75–85	263 (20.7)	135 (19.7)	128 (21.8)	
>85	707 (55.5)	370 (54.0)	337 (57.3)	
Female sex	448 (35.2)	243 (35.5)	205 (34.9)	0.820
Insurance status National health insurance only With Medicaid	707 (55.5) 566 (44.5)	379 (55.3) 306 (44.7)	328 (55.8) 260 (44.2)	0.871
Living conditions With family Veterans home Long-term care facilities Solitary living Others	998 (78.4) 53 (4.2) 134 (10.5) 59 (4.6) 29 (2.3)	543 (79.3) 28 (4.1) 64 (9.3) 34 (5.0) 16 (2.3)	455 (77.4) 25 (4.3) 70 (11.9) 25 (4.3) 13 (2.2)	0.644
Marital status Single Married Divorced Widow or widower	101 (7.9) 781 (61.4) 44 (3.5) 347 (27.3)	63 (9.2) 420 (61.3) 25 (3.6) 177 (25.8)	38 (6.5) 361 (61.4) 19 (3.2) 170 (28.9)	0.237
Religion Taoism Buddhism Catholic/Christian Others None	218 (17.1) 430 (33.8) 109 (8.6) 14 (1.1) 502 (39.4)	110 (16.1) 230 (33.6) 60 (8.8) 11 (1.6) 274 (40.0)	108 (18.4) 200 (34.0) 49 (8.3) 3 (0.5) 228 (38.8)	0.326
Educational level Higher than high school Lower than high school	555 (43.6) 718 (56.4)	303 (44.2) 382 (55.8)	252 (42.9) 336 (57.1)	0.621
Current alcohol consumption	14 (1.1)	9 (1.3)	5 (0.9)	0.475
Current smoker	68 (5.3)	44 (6.4)	24 (4.1)	0.116
TTAS * Emergent (triage 1) Urgent (triage 2) Non-urgent (triage 3, 4)	492 (38.6) 460 (36.1) 320 (25.1)	293 (42.8) 239 (34.9) 152 (22.2)	199 (33.8) 221 (37.6) 168 (28.6)	0.002
Glasgow Coma Scale	10.3 ± 4.4	10.2 ± 4.5	10.3 ± 4.3	0.717
13–15	536 (42.1)	296 (43.2)	240 (40.8)	0.035
5–12	543 (42.7)	272 (39.7)	271 (46.1)	
3–4	194 (15.2)	117 (17.1)	77 (13.1)	
Mean blood pressure in the emergency department (ED) (mmHg)	89.7 ± 23.7	89.4 ± 24.1	90.2 ± 23.3	0.565
Charlson Comorbidity Index	7.1 ± 2.4	7.1 ± 2.5	7.1 ± 2.3	0.672
≤3	37 (2.9)	20 (2.9)	17 (2.9)	0.888
4–6	545 (42.8)	289 (42.2)	256 (43.5)	
≥7	691 (54.3)	376 (54.9)	315 (53.6)	
APACHE II score at admission 0–14 15–24 >24	22.5 ± 8.3 216 (17.0) 546 (42.9) 511 (40.1)	22.5 ± 8.7 128 (18.7) 292 (42.6) 265 (38.7)	22.6 ± 7.7 88 (15.0) 254 (43.2) 246 (41.8)	0.834 0.184
Hospital length of stay (day)	21.5 ± 24.0	21.2 ± 26.6	21.7 ± 20.6	0.709
Total hospital expense (point)	293,627 ± 334,304	293,169 ± 350,043	294,161 ± 315,276	0.958
In-hospital mortality *	534 (41.9)	362 (52.8)	172 (29.3)	<0.001
DNR signed (Total)	1151 (90.4)	668 (97.5)	483 (82.1)	<0.001
DNR signed at admission	827 (65.0)	533 (77.8)	294 (50.0)	<0.001

The results are expressed as number (%) for categorical variables and mean ± standard deviation for numerical variables. TTAS = Taiwan Triage and Acuity Scale; ED = emergency department; APACHE = Acute Physiology and Chronic Health Evaluation; ICU = intensive care unit. * *p* < 0.05 is considered statistically significant using Mann–Whitney *U* test or chi-squared analysis.

**Table 3 ijerph-18-12546-t003:** Univariate and multiple logistic regression analysis of clinical characteristics between patients with ED-PC and UC.

Variable	Univariate Analysis			Multiple Logistic Regression		
	OR	95% CI	*p*	AOR	95% CI	*p*
A1	1.527	1.186–1.967	0.001	1.216	0.866–1.709	0.259
A6	0.699	0.561–0.873	0.002	0.899	0.682–1.185	0.451
A8 *	2.228	1.392–3.564	0.001	2.217	1.295–3.797	0.004
B2 *	1.388	1.112–1.731	0.004	1.348	1.040–1.748	0.024
B3 *	1.512	1.212–1.888	<0.001	1.696	1.315–2.187	<0.001
B6	0.612	0.430–0.870	0.006	0.800	0.534–1.198	0.278
B8	1.934	1.165–3.210	0.011	1.392	0.794–2.439	0.248
Age (y)	0.991	0.983–0.999	0.026	0.994	0.984–1.005	0.307
TTAS *			0.002			0.024
Emergent (triage 1)	1			1		
Urgent (triage 2)	0.734	0.568–0.949		0.763	0.575–1.013	
Non-urgent (triage 3, 4)	0.614	0.463–0.816		0.649	0.470–0.896	
Glasgow Coma Scale			0.035			0.658
13–15	1			1		
5–12	0.814	0.641–1.034		0.892	0.619–1.286	
3–4	1.232	0.882–1.721		1.010	0.679–1.502	
In-hospital mortality *	2.711	2.148–3.420	<0.001	1.983	1.540–2.555	<0.001
DNR signed (total)	8.542	5.050–14.449	<0.001	4.536	2.522–8.158	<0.001
DNR signed at admission	3.507	2.753–4.467	<0.001	2.133	1.619–2.811	<0.001

A1 = advanced cancer, metastatic, or locally aggressive disease; A6 = advanced central neurological diseases (e.g., stroke, dementia) (long-term bed-bound) combined with repeatedly or severely progressive deterioration or recurrent pneumonia, shortness of breath, or respiratory failure requiring hospital admission; A8 = very severely frail (completely dependent, approaching the end-of-life, CSHA-CFS > scale 8 and 9); B2 = appearance of progressive functional deterioration with ≥three ADLs requiring assistance; B3 = appearance of biopsychosocial discomforts requiring hospital admission; B6 = bed-bound patients with long-term unhealed bed sore or ulceration; B8 = patient’s family request for palliative care; TTAS = Taiwan Triage and Acuity Scale. OR = odds ratio; 95% CI = 95% confidence interval; AOR = adjusted odds ratio. * *p* < 0.05, considered statistically significant in the regression model.

**Table 4 ijerph-18-12546-t004:** Comparison of the end-of-life care in hospitalization between 362 ED-PC patients with mortality and 172 UC patients with mortality.

Variable	ED-PC *n* = 362 (%)	UC *n* = 172 (%)	*p*
Place of death * Intensive care unit Wards Hospice unit Home hospice	118 (32.6) 145 (40.1) 60 (16.6) 39 (10.8)	62 (36.0) 81 (47.1) 13 (7.6) 16 (9.3)	0.030
End-of-life care ET intubation	52 (14.4)	26 (15.1)	0.818
CPR	10 (2.8)	6 (3.5)	0.646
Epinephrine *	55 (15.2)	56 (32.6)	<0.001
Cardioversion or defibrillation	4 (1.1)	2 (1.2)	0.953
Vasopressors	221 (61.0)	114 (66.3)	0.243
Cardiac pacemaker	3 (0.8)	1 (0.6)	0.757
Ventilator support	57 (15.7)	28 (16.3)	0.875
ECMO or IABP	3 (0.8)	1 (0.6)	0.757
Withdrawal of ET tube *	16 (4.4)	1 (0.6)	0.018
Narcotics use *	223 (61.6)	76 (44.2)	<0.001
DNR signed (Total)	359 (99.2)	165 (95.9)	0.010
DNR signed at admission	285 (78.7)	124 (72.1)	0.091

Results expressed as number (%) for categorical variables. ET endotracheal; CPR cardiopulmonary resuscitation; ECMO extracorporeal membrane oxygenation; IABP intra-aortic balloon pump. * *p* < 0.05 is considered statistically significant using chi-square test or Fisher’s exact test.

**Table 5 ijerph-18-12546-t005:** Multiple logistic regression analyses of end-of-life care between 362 ED-PC patients with mortality and 172 UC patients with mortality.

Variable	Univariate Analysis			Multiple Logistic Regression		
	OR	95% CI	*p*	AOR	95% CI	*p*
Place of death			0.030			0.157
Intensive care unit	1			1		
Wards	0.941	0.624–1.418		0.731	0.467–1.143	
Hospice unit	2.425	1.236–4.757		1.487	0.723–3.058	
Home hospice	1.281	0.663–2.473		0.972	0.487–1.940	
Epinephrine *	0.371	0.242–0.570	<0.001	0.424	0.265–0.678	<0.001
Withdrawal of ET tube *	7.908	1.040–60.123	0.046	8.780	1.122–68.720	0.038
Narcotics use *	2.027	1.403–2.928	<0.001	1.675	1.132–2.477	0.010
DNR signed (Total)	5.077	1.296–19.879	0.020	2.572	0.622–10.634	0.192

ET = endotracheal; OR = odds ratio; 95% CI = 95% confidence interval. * *p* < 0.05 is considered statistically significant in the regression model.

## Data Availability

For available data please write to corresponding author David Hung-Tsang Yen, email: hjyen@vghtpe.gov.tw.

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
