# Peer review of "Differences in Characteristics, Hospital Care, and Outcomes between Acute Critically Ill Emergency Department Patients Receiving Palliative Care and Usual Care"

_ijerph, 2021, doi:10.3390/ijerph182312546_

Round 1

Reviewer 1 Report

It is clear that the authors of the work have thoroughly reviewed all parts of it.
There are no further comments to make, and the issue affecting the annex could not be reviewed, because it was not accessible on this occasion.

I trust the authors to review that question in the annex.

Reviewer 2 Report

Dear authors,

Thanks for the comments.

I believe the manuscript has been significantly improved and now warrants publication in IJERPH. 

This manuscript is a resubmission of an earlier submission. The following is a list of the peer review reports and author responses from that submission.

Round 1

Reviewer 1 Report

- In the Abstract (and in the paper as a whole) abbreviations and data are abused. The text cannot be read with agility or interest.
-The research Aim does not adequately express its purpose.
- On line 21 there is a double space.
- In lines 28 and 29 there are spaces inside the parentheses that should not be like this.
The Keywords do not understand the use of "Do-not-resuscitate". In fact the term does not appear in some databases consulted.
The Introduction seems excessively short and little argued. References 2 and 3 are cited continuously and incorrectly in the text.
Table 1 of the results is redundant with lines 107 to 127.
I think that the interpretation of the data should be reviewed since comparing the percentages in relation to the number of patients who agreed to participate in ED-PC, with the 588 who received UC, I am not sure if it is correct.
In line 130 it appears "Of the 1273 patients with unmet palliative care, 52.8% (672/1273) had one item, 43.8% (558/1273) ...” it is not understood which item the author refers to.
Table 1 within the text, has in the fourth column and a p-value located differently from the table in the annex.
Figures 1 and 2 do not have the same format, at least typeface, as the rest of the article.
The data presented in the Results and the Discussion of them is disproportionately extensive in relation to the Introduction, Aim and Conclusions.
References should be thoroughly reviewed as they do not respect a single style.

Author Response

Reviewer 1

Q: In the Abstract (and in the paper as a whole) abbreviations and data are abused. The text cannot be read with agility or interest.

A: We are grateful of reviewer’s thoughtful comment and decreased the use of abbreviations significantly in abstract and all throughout the context of the paper.

Q: The research Aim does not adequately express its purpose.

A: We are grateful of reviewer’s thoughtful comments. Since our aim was simple and hard to augment and embellish on the aim itself we modified add more arguments to the introduction.

Q: On line 21 there is a double space.

A: We are grateful for reviewer’s thoughtful comment, we modified it to single space.

Q: In lines 28 and 29 there are spaces inside the parentheses that should not be like this.

A: We are grateful for reviewer’s thoughtful comment, we modified it accordingly.

Q: The Keywords do not understand the use of "Do-not-resuscitate". In fact the term does not appear in some databases consulted.

A: We agreed with reviewer’s thoughtful comment. In this paper we did not discuss DNR as readily as in our previous papers. Hence we deleted the keywords.

Q: The Introduction seems excessively short and little argued.

A: We are grateful to the reviewer’s thoughtful comments and modified and added to the introduction accordingly.

Q: References 2 and 3 are cited continuously and incorrectly in the text.

A: We are grateful to the reviewer’s thoughtful comment and modified the text by deleting text with citing of references 2 and 3 in discussion and some part of introduction. In fact, we removed the citing of reference 3 all together from the paper.

Q: Table 1 of the results is redundant with lines 107 to 127.

A: We agree with reviewer’s thoughtful comments and deleted the redundant parts.

Q: I think that the interpretation of the data should be reviewed since comparing the percentages in relation to the number of patients who agreed to participate in ED-PC, with the 588 who received UC, I am not sure if it is correct.

A: We are grateful for the reviewer’s thoughtful comments. We reviewed the data, the percentages in relation to the number of patients in ED-PC and UC are in agreement in the study and in the text. What may cause confusion is table 4 where we identify only patients with mortality (362 ED-PC patients with mortality and 172 UC patients with mortality) to compare the difference in end-of-life care.

Q: In line 130 it appears "Of the 1273 patients with unmet palliative care, 52.8% (672/1273) had one item, 43.8% (558/1273) ...” it is not understood which item the author refers to.

A: We are grateful to reviewer’s thoughtful comments. The text was ambiguous and not contributing to the clarity of the manuscript, hence we deleted the text.

Q: Table 1 within the text, has in the fourth column and a p-value located differently from the table in the annex.

A: Dear reviewer, we are confused by the question. Please clarify so we can modify accordingly.

Q: Figures 1 and 2 do not have the same format, at least typeface, as the rest of the article.

A: We are grateful for the reviewer’s thoughtful comments, we changed the typeface to Palatino Linotype to match the context of the manuscript.

Q: The data presented in the Results and the Discussion of them is disproportionately extensive in relation to the IntroductionAim and Conclusions.

A: We agree with reviewer’s thoughtful comments and added to introduction and deleted some parts of results and discussion.

Q: References should be thoroughly reviewed as they do not respect a single style.

A: We are grateful to the reviewer’s thoughtful comments, we modified the references in a consistent way accordingly in the reference section.

Reviewer 2 Report

Dear Author/s,

The topic is interesting and important as it emphasizes the importance of early involvement of palliative care. The overall manuscript is clearly written and well organized.

I suggested to use only one way of citing references, according to the guidelines for authors.

Author Response

Reviewer 2

The topic is interesting and important as it emphasizes the importance of early involvement of palliative care. The overall manuscript is clearly written and well organized.

Q: I suggested to use only one way of citing references, according to the guidelines for authors.

A: We are grateful to the reviewer’s thoughtful comments, we modified the references in a consistent way accordingly in the reference section.

Reviewer 3 Report

Thanks for the opportunity to review this interesting paper. 

Palliative care constitutes a health domain that must be addressed and discussed amongst health professionals, as a multidisciplinary team. 

Although that, I suggest authors improve the introduction providing sufficient theoretical background and include all relevant references. Also, it is not clear the reason to have conducted a retrospective observational analysis in the ED of a tertiary medical centre - the Taipei Veterans General Hospital (TVGH)-  and why the Institutional Research Board waived the need for patient consent. 

In the abstract authors should improve clarity, turning the abstract more comprehensive and highlighting the novelty of the study, its main goals and findings.

Hope these comments contribute to the paper's improvement.

Author Response

Reviewer 3

Q: I suggest authors improve the introduction providing sufficient theoretical background and include all relevant references.

A: We are grateful for the reviewer’s thoughtful comment and added to the introduction accordingly.

Q: it is not clear the reason to have conducted a retrospective observational analysis in the ED of a tertiary medical centre - the Taipei Veterans General Hospital (TVGH)

A: We are grateful for the reviewer’s thoughtful comment and added to the introduction clarity why we conducted a retrospective observation in the ED.

Q: why the Institutional Research Board waived the need for patient consent. 

A: We are grateful for the author’s thoughtful comment. Since it is a retrospective study, therefore the needs for informed consent was waived.

Q: In the abstract authors should improve clarity, turning the abstract more comprehensive and highlighting the novelty of the study, its main goals and findings.

A: We are thoughtful of reviewer’s comment and modified abstract accordingly.
